# Practical Adversarial Attacks on Spatiotemporal Traffic Forecasting Models

**Fan LIU**
AI Thrust&RBM, The Hong Kong University of Science and Technology (Guangzhou)
`fliu236@connect.hkust-gz.edu.cn&liufan@ust.hk`

**Hao LIU**[*]
AI Thrust, The Hong Kong University of Science and Technology (Guangzhou)
Guangzhou HKUST Fok Ying Tung Research Institute
CSE, The Hong Kong University of Science and Technology
`liuh@ust.hk`

**Wenzhao Jiang**
AI Thrust, The Hong Kong University of Science and Technology (Guangzhou)
`wjiang431@connect.hkust-gz.edu.cn`

## Abstract

Machine learning based traffic forecasting models leverage sophisticated spatiotemporal auto-correlations to provide accurate predictions of city-wide traffic states. However, existing methods assume a reliable and unbiased forecasting environment, which is not always available in the wild. In this work, we investigate the vulnerability of spatiotemporal traffic forecasting models and propose a practical adversarial spatiotemporal attack framework. Specifically, instead of simultaneously attacking all geo-distributed data sources, an iterative gradient-guided node saliency method is proposed to identify the time-dependent set of victim nodes. Furthermore, we devise a spatiotemporal gradient descent based scheme to generate real-valued adversarial traffic states under a perturbation constraint. Meanwhile, we theoretically demonstrate the worst performance bound of adversarial traffic forecasting attacks. Extensive experiments on two real-world datasets show that the proposed two-step framework achieves up to $67.8\%$ performance degradation on various advanced spatiotemporal forecasting models. Remarkably, we also show that adversarial training with our proposed attacks can significantly improve the robustness of spatiotemporal traffic forecasting models. Our code is available in `https://github.com/usail-hkust/Adv-ST`.

## 1 Introduction

Machine learned spatiotemporal forecasting models have been widely adopted in modern Intelligent Transportation Systems (ITS) to provide accurate and timely prediction of traffic dynamics, *e.g.*, traffic flow [1], traffic speed [2, 3], and the estimated time of arrival [4, 5]. Despite fruitful progress in improving the forecasting accuracy and utility [6], little attention has been paid to the robustness of spatiotemporal forecasting models. For example, Figure 1 demonstrates that injecting slight adversarial perturbations on a few randomly selected nodes can significantly degrade the traffic

---

[*]Corresponding author

36th Conference on Neural Information Processing Systems (NeurIPS 2022).

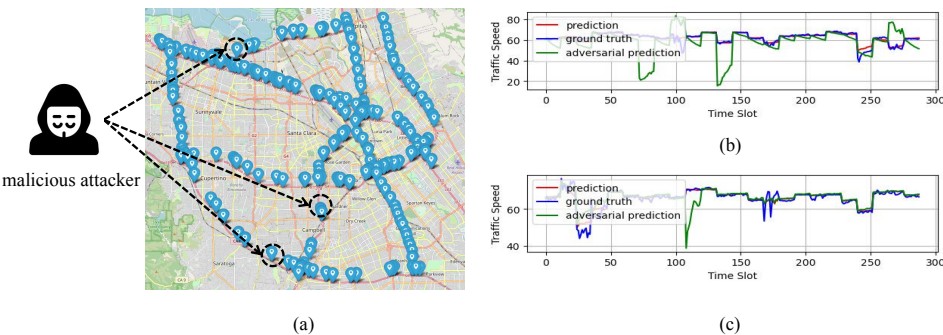

Figure 1: An illustration of adversarial attack against spatiotemporal forecasting models on the Bay Area traffic network in California, the data ranges from January 2017 to May 2017. (a) Adversarial attack of geo-distributed data. The malicious attacker may inject adversarial examples into a few randomly selected geo-distributed data sources. (*e.g.*, roadway sensors) to mislead the prediction of the whole traffic forecasting system. (b) Accuracy drop of victim nodes. By adding less than 50% traffic speed perturbations to 10% victim nodes, we observe 60.4% accuracy drop of victim nodes in morning peak hour. (c) Accuracy drop of neighbouring nodes. Due to the information diffusion of spatiotemporal forecasting models, the adversarial attack also leads to up to about 47.23% accuracy drop for neighboring nodes.

forecasting accuracy of the whole system. Therefore, this paper investigates the vulnerability of traffic forecasting models against adversarial attacks.

In recent years, adversarial attacks have been extensively studied in various application domains, such as computer vision and natural language processing [7]. However, two major challenges prevent applying existing adversarial attack strategies to spatiotemporal traffic forecasting. First, the traffic forecasting system makes predictions by exploiting signals from geo-distributed data sources (*e.g.*, hundreds of roadway sensors and thousands of in-vehicle GPS devices). It is expensive and impractical to manipulate all data sources to inject adversarial perturbations simultaneously. Furthermore, state-of-the-art traffic forecasting models propagate local traffic states through the traffic network for more accurate prediction [5]. Attacking a few arbitrary data sources will result in node-varying effects on the whole system. How to identify the subset of salient victim nodes with a limited attack budget to maximize the attack effect is the first challenge. Second, unlike most existing adversarial attack strategies that focus on time-invariant label classification [8, 9], the adversarial attack against traffic forecasting aims to disrupt the target model to make biased predictions of continuous traffic states. How to generate real-valued adversarial examples without access to the ground truth of future traffic states is another challenge.

To this end, in this paper, we propose a practical adversarial spatiotemporal attack framework that can disrupt the forecasting models to derive biased city-wide traffic predictions. Specifically, we first devise an iterative gradient-guided method to estimate node saliency, which helps to identify a small time-dependent set of victim nodes. Moreover, a spatiotemporal gradient descent scheme is proposed to guide the attack direction and generate real-valued adversarial traffic states under a human imperceptible perturbation constraint. The proposed attack framework is agnostic to forecasting model architecture and is generalizable to various attack settings, *i.e.*, white-box attack, grey-box attack, and black-box attack. Meanwhile, we theoretically analyze the worst performance guarantees of adversarial traffic forecasting attacks. We prove the adversarial robustness of spatiotemporal traffic forecasting models is related to the number of victim nodes, the maximum perturbation bound, and the maximum degree of the traffic network.

Extensive experimental studies on two real-world traffic datasets demonstrate the attack effectiveness of the proposed framework on state-of-the-art spatiotemporal forecasting models. We show that attacking $10\%$ nodes in the traffic system can break down the global forecasting Mean Average Error (MAE) from $1.975$ to $6.1329$. Moreover, the adversarial attack can induce $68.65\%$, and $56.67\%$ performance degradation under the extended white-box and black-box attack settings, respectively. Finally, we also show that incorporating adversarial examples we generated with adversarial training can significantly improve the robustness of spatiotemporal traffic forecasting models.

## 2 Background and problem statement

In this section, we first introduce some basics of spatiotemporal traffic forecasting and adversarial attack, then formally define the problem we aim to address.

### 2.1 Spatiotemporal traffic forecasting

Let $\mathcal{G}_t = (\mathcal{V}, \mathcal{E})$ denote a traffic network at time step $t$, where $\mathcal{V}$ is a set of $n$ nodes (*e.g.*, regions, road segments, roadway sensors, *etc.*) and $\mathcal{E}$ is a set of edges. The construction of $\mathcal{G}_t$ can be categorized into two types, (1) prior-based, which pre-define $\mathcal{G}_t$ based on metrics such as geographical proximity and similarity [10], and (2) learning-based, which automatically learns $\mathcal{G}_t$ in an end-to-end way [2]. Note the $\mathcal{G}_t$ can be static or time-evolving depending on the forecasting model. We denote $\mathbf{X}_t = (\mathbf{x}_{1,t}, \mathbf{x}_{2,t}, \cdots, \mathbf{x}_{n,t})$ as the spatiotemporal features associated to $\mathcal{G}_t$, where $\mathbf{x}_{i,t} \in \mathbb{R}^c$ represents the $c$-dimensional time-varying traffic conditions (*e.g.*, traffic volume, traffic speed) and contextual features (*e.g.*, weather, surrounding POIs) of node $v_i \in \mathcal{V}$ at $t$. The spatiotemporal traffic forecasting problem aims to predict traffic states for all $v_i \in \mathcal{V}$ over the next $\tau$ time steps,

$$\hat{\mathbf{Y}}_{t+1:t+\tau} = f_\theta(\mathcal{H}_{t-T+1:t}), \tag{1}$$

where $\mathcal{H}_{t-T+1:t} = \{(\mathbf{X}_{t-T+1}, \mathcal{G}_{t-T+1}), \ldots, (\mathbf{X}_t, \mathcal{G}_t)\}$ denotes the traffic states contains input features and the traffic network in previous $T$ time steps, $f_\theta(\cdot)$ is the spatiotemporal traffic forecasting model parameterized by $\theta$, and $\hat{\mathbf{Y}}_{t+1:t+\tau} = \{\hat{\mathbf{Y}}_{t+1}, \hat{\mathbf{Y}}_{t+2}, \cdots, \hat{\mathbf{Y}}_{t+\tau}\}$ is the estimated traffic conditions of interest of $\mathcal{V}$ from time step $t+1$ to $t+\tau$. We denote $\mathbf{Y}_{t+1:t+\tau} = \{\mathbf{Y}_{t+1}, \mathbf{Y}_{t+2}, \cdots, \mathbf{Y}_{t+\tau}\}$ as the ground truth of $\mathcal{H}_{t-T+1:t}$.

Note the above formulation is consistent with the state-of-the-art Graph Neural Network (GNN) based spatiotemporal traffic forecasting models [2, 10, 11, 12], and is also generalizable to other variants such as Convolutional Neural Network (CNN) based approaches [13].

### 2.2 Adversarial attack

Given a machine learning model, adversarial attack aims to mislead the model to derive biased predictions by generating the optimal adversarial example

$$x^* \in \arg\max_{x'} \mathcal{L}(x', y; \theta) \quad s.t. \|x' - x\|_p \leq \varepsilon, \tag{2}$$

where $x'$ is the adversarial example with maximum bound $\varepsilon$ under $L_p$ norm to guarantee the perturbation is imperceptible to human, and $y$ is the ground truth of clean example $x$. Various gradient-based methods have been proposed to generate adversarial examples, such as FGSM [14], PGD [8], MIM [9], *etc.* For instance, the adversarial example $x' = x + \varepsilon \text{sign}(\nabla_x \mathcal{L}_{CE}(x, y; \theta))$ in FGSM, where $\text{sign}(\cdot)$ is the Signum function and $\mathcal{L}_{CE}(\cdot)$ is the cross entropy loss.

Note the adversarial attack happened in the testing stage, and the attackers cannot manipulate the forecasting model or its output. On the benign testing set, the forecasting model can perform well. Based on the amount of information the attacker can access in the testing stage, the adversarial attack can be categorized into three classes. *White-box attack*. The attacker can fully access the target model, including the model architecture, the model parameters, gradients, model outputs, the input traffic states, and the corresponding labels. *Grey-box attack*. The attacker can partially access the system, including the target model and the input traffic states, but without the labels. *Black-box attack*. The attacker can only access the input traffic states, query the outputs of the target model or leverage a surrogate model to craft the adversarial examples.

### 2.3 Adversarial attack against spatiotemporal traffic forecasting

This work aims to apply adversarial attacks to spatiotemporal traffic forecasting models. We first define the *adversarial traffic state* as follow,

$$\mathcal{H}'_t = \left\{ (\mathbf{X}'_t, \mathcal{G}_t) : \|S_t\|_0 \leq \eta, \|(\mathbf{X}'_t - \mathbf{X}_t) \cdot S_t\|_p \leq \varepsilon \right\}, \tag{3}$$

where $S_t \in \{0, 1\}^{n \times n}$ is a diagonal matrix with $i$th diagonal element indicating whether node $i$ is a victim node, and $\mathbf{X}'_t$ is the perturbed spatiotemporal feature named adversarial spatiotemporal feature.

We restrict the adversarial traffic state by the victim node budget $\eta$ and the perturbation budget $\varepsilon$. Note that following the definition of adversarial attack, we leave the topology of $\mathcal{G}_t$ immutable as we regard the adjacency relationship as a part of the model parameter that may be automatically learned in an end-to-end way.

*Attack goal.* The attacker aims to craft adversarial traffic states to fool the spatiotemporal forecasting model to derive biased predictions. Formally, given a spatiotemporal forecasting model $f_\theta(\cdot)$, the adversarial attack against spatiotemporal traffic forecasting is defined as

$$\max_{\substack{\mathcal{H}'_{t-T+1:t} \\ t \in \mathcal{T}_{test}}} \sum_{t \in \mathcal{T}_{test}} \mathcal{L}(f_{\theta^*}(\mathcal{H}'_{t-T+1:t}), \mathbf{Y}_{t+1:t+\tau}) \tag{4a}$$

$$s.t., \quad \theta^* = \arg\min_\theta \sum_{t \in \mathcal{T}_{train}} \mathcal{L}(f_\theta(\mathcal{H}_{t-T+1:t}), \mathbf{Y}_{t+1:t+\tau}), \tag{4b}$$

where $\mathcal{T}_{test}$ and $\mathcal{T}_{train}$ denote the set of time steps of all testing and training samples, respectively. $\mathcal{L}(\cdot)$ is the loss function measuring the distance between the predicted traffic states and ground truth, and $\theta^*$ is optimal parameters learned during the training stage.

Since the ground truth (*i.e.*, future traffic states) under the spatiotemporal traffic forecasting setting is unavailable at run-time, the practical adversarial spatiotemporal attack primarily falls into the grey-box attack setting. However, investigating white-box attacks is still beneficial to help us understand how adversarial attack works and can help improve the robustness of spatiotemporal traffic forecasting models (*e.g.*, apply adversarial training). We discuss how to extend our proposed adversarial attack framework to white-box and black-box settings in Section 3.2.

# 3 Methodology

In this section, we introduce the practical adversarial spatiotemporal attack framework in detail. Specifically, our framework consists of two steps: (1) identify the time-dependent victim nodes, and (2) attack with the adversarial traffic state.

## 3.1 Identify time-dependent victim nodes

One unique characteristic that distinguishes attacking spatiotemporal forecasting from conventional classification tasks is the inaccessibility of ground truth at the test phase. Therefore, we first construct future traffic states' surrogate label to guide the attack direction,

$$\tilde{\mathbf{Y}}_{t+1:t+\tau} = g_\phi(\mathcal{H}_{t-T+1:t}) + \delta_{t+1:t+\tau}, \tag{5}$$

where $g_\phi(\cdot)$ is a generalized function (*e.g.*, $\tanh(\cdot)$, $\sin(\cdot)$, $f_\theta(\cdot)$), $\delta_{t+1:t+\tau}$ are random variables sampled from a probability distribution $\pi(\delta_{t+1:t+\tau})$ to increase the diversity of the attack direction. In our implementation, we derive $\phi$ based on the pre-trained forecasting model parameter $\theta^*$, and $\delta_{t+1:t+\tau} \sim U(-\varepsilon/10, \varepsilon/10)$. In the real-world production [5], the forecasting models are usually updated in an online fashion (*e.g.*, per hours). Therefore, we estimate the missing latest traffic states based on previous input data, $\tilde{\mathcal{H}}_t = g_\varphi(\mathcal{H}_{t-1})$, where $g_\varphi(\cdot)$ is the estimation function parameterized by $\varphi$. For simplicity, we directly obtain $\varphi$ from the pre-trained traffic forecasting model $f_{\theta^*}(\cdot)$.

With the surrogate traffic state label $\tilde{\mathbf{Y}}_{t+1:t+\tau}$, we derive the time-dependent node saliency (TDNS) for each node as

$$\mathcal{M}_t = \left\| \sigma\left(\frac{\partial \mathcal{L}(f_\theta(\tilde{\mathcal{H}}_{t-T+1:t}), \tilde{\mathbf{Y}}_{t+1:t+\tau})}{\partial \tilde{\mathbf{X}}_{t-T+1:t}}\right) \right\|_p, \tag{6}$$

where $\mathcal{L}(f_\theta(\tilde{\mathcal{H}}_{t-T+1:t}), \tilde{\mathbf{Y}}_{t+1:t+\tau})$ is the loss function and $\sigma$ is the activation function. Intuitively, $\mathcal{M}_t$ reveals the node-wise loss impact with the same degree of perturbations. Note depending on the time step $t$, $\mathcal{M}_t$ may vary. A similar idea also has been adopted to identify static pixel saliency for image classification [15].

More in detail, the loss function $\mathcal{L}(f_\theta(\tilde{\mathcal{H}}_{t-T+1:t}), \tilde{\mathbf{Y}}_{t+1:t+\tau})$ in Equation 6 is updated by the iterative gradient-based adversarial method [8],

$$\mathbf{X}'^{(i)}_{t-T+1:t} = \mathrm{clip}_{\mathbf{X}'_{t-T+1:t}, \varepsilon}(\mathbf{X}'^{(i-1)}_{t-T+1:t} + \alpha \mathrm{sign}(\nabla \mathcal{L}(f_{\theta^*}(\mathcal{H}'^{(i-1)}_{t-T+1:t}), \tilde{\mathbf{Y}}_{t+1:t+\tau}))), \tag{7}$$

where $\mathcal{H}'^{(i)}_{t-T+1:t}$ is adversarial traffic states at $i$-th iteration, $\alpha$ is the step size, and $\text{clip}_{\mathbf{X}'_{t-T+1,\varepsilon}}(\cdot)$ is the project operation which clips the spatiotemporal feature with maximum perturbation bound $\varepsilon$. Note $\mathcal{H}'^{(0)}_{t-T+1:t} = \tilde{\mathcal{H}}_{t-T+1:t}$.

For each batch of data $\left\{ (\tilde{\mathcal{H}}_{t-T+1:t}, \tilde{\mathbf{Y}}_{t+1:t+\tau})_{(j)} \right\}_{j=1}^{\gamma}$, the time-dependent node saliency gradient is derived by

$$\mathbf{g}_t = \frac{1}{\gamma} \sum_j \{ \frac{\partial \mathcal{L}(f_{\theta^*}(\tilde{\mathcal{H}}_{t-T+1:t}), \tilde{\mathbf{Y}}_{t+1:t+\tau})}{\partial \mathbf{X}'_{t-T+1:t}} \}_j, \tag{8}$$

where $\gamma$ is the batch size. We use the RELU activation function to compute the non-negative saliency score for each time step,

$$\mathcal{M}_t = \| \text{Relu}(\mathbf{g}_t) \|_2. \tag{9}$$

Finally, we obtain the set of victim node $S_t$ based on $\mathcal{M}_t$,

$$s_{(i,i),t} = \begin{cases} 1 & \text{if } v_i \in \text{Top}(\mathcal{M}_t, k) \\ 0 & \text{otherwise}, \end{cases} \tag{10}$$

where $s_{(i,i),t}$ denotes the $i$-th diagonal element of $S_t$, and $\text{Top}(\cdot)$ is a 0-1 indicator function returning if $v_i$ is the top-$k$ salient node at time step $t$.

## 3.2 Attack with adversarial traffic state

Based on the time-dependent victim set, we conduct adversarial attacks to spatiotemporal traffic forecasting models. Specifically, we first generate perturbed adversarial traffic features based on gradient descent methods. Take the widely used Projected Gradient Descent (PGD) [8] for illustration, we construct Spatiotemporal Projected Gradient Descent (STPGD) as below,

$$\mathbf{X}'^{(i)}_{t-T+1:t} = \text{clip}_{\mathbf{X}'_{t-T+1:t,\varepsilon}}(\mathbf{X}'^{(i-1)}_{t-T+1:t} + \alpha \text{sign}(\nabla \mathcal{L}(f_{\theta^*}(\mathcal{H}'^{(i-1)}_{t-T+1:t}), \tilde{\mathbf{Y}}_{t+1:t+\tau}) \cdot S_t)), \tag{11}$$

where $\mathcal{H}'^{(i-1)}_{t-T+1:t}$ is the adversarial traffic state at $i-1$-th iteration in the iterative gradient descent, $\alpha$ is the step size, and $\text{clip}_{\mathbf{X}'_{t-T+1:t,\varepsilon}}(\cdot)$ is the operation to bound adversarial features in a $\varepsilon$ ball. Note $\mathbf{X}'^{(0)}_t = \tilde{\mathbf{X}}_t$. Instead of perturbing all nodes as in vanilla PGD, we only inject perturbations on selected victim nodes in $S_t$. Similarly, we can generate perturbed adversarial traffic features by extending other gradient based methods, such as MIM [9].

In the testing phase, we can inject the adversarial traffic states $\mathcal{H}'_{t-T+1:t} = \mathcal{H}_{t-T+1:t} + \triangle \mathcal{H}'_{t-T+1:t}$ to apply adversarial attack, where $\triangle \mathcal{H}'_t + \mathcal{H}_t = \{ (\mathbf{X}'_t - \mathbf{X}_t) \cdot S_t + \mathbf{X}_t, \mathcal{G}_t \} \in \mathcal{H}'_{t-T+1:t}$ and $\triangle \mathcal{H}'_t = \left\{ ((\mathbf{X}'_t - \mathbf{X}_t) \cdot S_t, 0) : \| S_t \|_0 \leq \eta, \| (\mathbf{X}'_t - \mathbf{X}_t) \cdot S_t \|_p \leq \varepsilon \right\} \in \triangle \mathcal{H}'_{t-T+1:t}$. The details of the adversarial spatiotemporal attack framework under the grey-box setting is in algorithm 1.

The overall adversarial spatiotemporal attack can be easily extended to the white-box and black-box settings, which are detailed below.

**White-box attack**. Since the adversaries can fully access the data and labels under the white-box setting, we directly use the real ground truth traffic states to guide the generation of adversarial traffic states. The detailed algorithm is introduced in Appendix A.1.

**Black-box attack**. The most restrictive black-box setting assumes limited accessibility to the target model and labels. Therefore, we first employ a surrogate model, which can be learned from the training data or by querying the traffic forecasting service [16, 17]. Then we generate adversarial traffic states based on the surrogate model to attack the targeted traffic forecasting model. Please refer to Appendix A.2 for more details.

We conclude this section with the theoretical upper bound analysis of the proposed adversarial attack strategy. In particular, we demonstrate the attack performance against the spatiotemporal traffic forecasting model is related to the number of chosen victim nodes, the budget of adversarial perturbations, as well as the traffic network topology.

**Theorem 1** *Let $\mathbf{Z}^{(L)} = f_\theta(\mathcal{H}_{t-T+1:t})$ and $\mathbf{Z}'^{(L)} = f_\theta(\mathcal{H}'_{t-T+1:t})$ be the L-th layer embeddings of the forecasting model, the upper bound of the adversarial loss satisfies*

$$\left\| \mathbf{Z}^{(L)} - \mathbf{Z}'^{(L)} \right\|_2^2 \leq (\lambda\beta C)^{2L}\varepsilon^2\eta,$$

*where $\lambda$ denotes maximum weight bound in all layers of the forecasting model, $\beta$ denotes parameter of the activation function in $f_\theta(\cdot)$, $C$ denotes the maximum degree of $\mathcal{G}$. $\eta$ and $\varepsilon$ are the budget of number of victim nodes and perturbations, respectively.*

*Proof.* Please refer to Appendix B.

---

**Algorithm 1:** Adversarial spatiotemporal attack under the grey-box setting

---

**Input:** Previous traffic data, pre-trained spatiotemporal model $f_{\theta^*}(\cdot)$, pre-trained traffic state prediction model $g_\varphi(\cdot)$, maximum perturbation budget $\varepsilon$, victim node budget $\eta$, and iterations $K$.

**Result:** Perturbed Adversarial traffic states $\mathcal{H}'_{t-T+1:t}$.

/\* Step 1:  Identify time-dependent victim nodes              \*/

1 Estimate current traffic state $\tilde{\mathcal{H}}_{t-N+1:t}$ by function $g_\varphi(\cdot)$;

2 Construct future traffic state's surrogate labels $\tilde{\mathbf{Y}}_{t+1:t+\tau}$ by Equation 5 ;

3 Compute the time-dependent node saliency $\mathcal{M}_t$ with $\tilde{\mathcal{H}}_{t-T+1:t}$ and $\tilde{\mathbf{Y}}_{t+1:t+\tau}$ by Equation 6-9;

4 Obtain the victim node set $S_t$ by Equations 10 ;

/\* Step 2:  Attack with adversarial traffic state              \*/

5 Initialize adversarial traffic state $\mathcal{H}'^{(0)}_{t-T+1:t} = \tilde{\mathcal{H}}_{t-T+1:t}$;

6 **for** $i = 1$ *to* $K$ **do**

7      Generate perturbed adversarial features $\mathbf{X}'^{(i)}_{t-T+1:t}$ by Equation 11;

8      $\triangle\mathcal{H}'^{(i)}_{t-T+1:t} = ((\mathbf{X}'^{(i)}_{t-T+1:t} - \tilde{\mathbf{X}}_{t-T+1:t}) \cdot S_t, 0)$;

9 **end**

10 Return $\mathcal{H}'_{t-T+1:t} = \mathcal{H}_{t-T+1:t} + \triangle\mathcal{H}'_{t-T+1:t}$.

---

# 4 Experiments

## 4.1 Experimental setup

**Datasets**. We use two popular real-world datasets to demonstrate the effectiveness of the proposed adversarial attack framework. (1) *PEMS-BAY* [18] traffic dataset is derived from the California Transportation Agencies (CalTrans) Performance Measurement System (PeMS) ranging from January 1, 2017 to May 31, 2017. 325 traffic sensors in the Bay Area collect traffic data every 5 minutes. (2) *METR-LA* [19] is a traffic speed dataset collected from 207 Los Angeles County roadway sensors. The traffic speed is recorded every 5 minutes and ranges from March 1, 2012 to June 30, 2012. For evaluation, all datasets are chronologically ordered, we take the first 70% for training, the following 10% for validation, and the rest 20% for testing. The statistics of the two datasets are reported in Appendix C.

**Baselines**. In the current literature, few studies can be directly applied to the real-valued traffic forecasting attack setting. To guarantee the fairness of comparison, we construct two-step baselines as below. For victim node identification, we adopt random selection and use the topology-based methods (*i.e.*, node degree and betweenness centrality [20]) to select victim nodes. We also employ PageRank (PR) [21] as the baseline to decide the set of victim nodes. For adversarial traffic state generation, we adopt two widely used iterative gradient-based methods, PGD [8] and MIM [9], to generate adversarial perturbations. In summary, we construct eight two-step baselines, PGD-Random, PGD-PR, PGD-Centrality, PGD-Degree, MIM-Random, MIM-PR, MIM-Centrality, and MIM-Degree. For instance, PGD-PR indicates first identifying victim nodes with PageRank and then applying adversarial noises with PGD. Depending on the adversarial perturbation method, we compare two variants of our proposed framework, namely STPGD-TDNS and STMIM-TDNS.

Table 1: Adversarial attack performance under the grey-box setting.

| Metrics / Attack methods | PeMS-BAY | | | | METR-LA | | | |
|---|---|---|---|---|---|---|---|---|
| | G-MAE | L-MAE | G-RMSE | L-RMSE | G-MAE | L-MAE | G-RMSE | L-RMSE |
| non-attack | 1.975 | - | 4.0220 | - | 6.3504 | - | 11.8424 | - |
| PGD-Random | 4.9876 | 3.7431 | 8.9343 | 7.8006 | 7.8947 | 2.7030 | 13.2749 | 5.9501 |
| PGD-PR | 4.8599 | 3.5819 | 8.8215 | 7.6727 | 7.9003 | 2.7070 | 13.2669 | 5.9132 |
| PGD-Centrality | 5.1640 | 3.9585 | 9.1369 | 8.0333 | 7.8554 | 2.7107 | 13.3100 | 5.9422 |
| PGD-Degree | 4.9121 | 3.6675 | 8.8486 | 7.7263 | 7.9011 | 2.7316 | 13.3738 | 6.0661 |
| MIM-Random | 5.3645 | 4.1739 | 9.7082 | 8.6825 | 7.7115 | 2.3793 | 13.1724 | 5.6882 |
| MIM-PR | 5.2405 | 4.0286 | 9.5902 | 8.5600 | 7.7206 | 2.3774 | 13.1294 | 5.6548 |
| MIM-Centrality | 5.5321 | 4.3820 | 9.9312 | 8.9331 | 7.7074 | 2.4255 | 13.2233 | 5.7498 |
| MIM-Degree | 5.3500 | 4.1745 | 9.5808 | 8.5573 | 7.7026 | 2.3877 | 13.2570 | 5.8229 |
| **STPGD-TDNS** | **6.1329** | **5.1647** | **10.6723** | **9.7003** | 7.7191 | 2.6534 | 13.6693 | 6.6794 |
| **STMIM-TDNS** | 5.6706 | 4.7010 | 10.1336 | 9.1813 | **7.9381** | **2.8848** | **13.8592** | **6.9885** |

**Target model**. To evaluate the generalization ability of the proposed adversarial attack framework, we adopt the state-of-the-art spatiotemporal traffic forecasting model, GraphWaveNet (Gwnet) [2], as the target model. Evaluation results on more target models are reported in Appendix F.

**Evaluation metrics**. Our evaluation focus on both the global and local effect of adversarial attacks on spatiotemporal models,

$$\mathbb{E}_{t \in \mathcal{T}_{test}} \mathcal{L}(f_\theta(\mathcal{H}'_{t-T+1:t}), \mathbf{Y}_{t+1:t+\tau}), \tag{12a}$$

$$\mathbb{E}_{t \in \mathcal{T}_{test}} \mathcal{L}(f_\theta(\mathcal{H}'_{t-T+1:t}), f_\theta(\mathcal{H}_{t-T+1:t})), \tag{12b}$$

where $\mathcal{L}(\cdot)$ is a user-defined loss function. Different from the majority target of adversarial attacks that are classification models (*e.g.*, adversarial accuracy), traffic forecasting is defined as a regression task. Therefore, we adopt Mean Average Error (MAE) [22] and Root Mean Square Error (RMSE) [23] for evaluation. More specifically, we define Global MAE (G-MAE), Local MAE (L-MAE), Global RMSE (G-RMSE), Local RMSE (L-RMSE) to evaluate the effect of adversarial attacks on traffic forecasting. Please refer to Appendix D for detailed definitions of four metrics.

**Implementation details**. All experiments are implemented with PyTorch and performed on a Linux server with 4 RTX 3090 GPUs.The traffic speed is normalized to $[0, 1]$. The input length $T$ and output length $\tau$ are set to 12. We select 10% nodes from the whole nodes as the victim nodes, and $\varepsilon$ is set to 0.5. The batch size $\gamma$ is set to 64. The iteration $K$ is set to 5, and the step size $\alpha$ is set to 0.1.

## 4.2 Overall attack performance

Table 1 reports the overall attack performance of our proposed approach against the original forecasting model and eight baselines with respect to four metrics. Note larger value indicates better attack performance and worse forecasting accuracy. Specifically, we can make the following observations. First, the adversarial attack can significantly degrade the traffic forecasting performance. For example, our approach achieves $(67.79\%, 62.31\%)$ and $(19.88\%, 14.55\%)$ global performance degradation compared with the original forecasting results on *PeMS-BAY* and *METR-LA* dataset, respectively. Second, our approach achieves the best attack performance against all baselines. In particular, STPGD-TDNS achieves $(15.80\%, 15.39\%)$ global performance improvement and $(23.35\%, 17.19\%)$ local performance improvement on the *PeMS-BAY* dataset. Similarly, STMIM-TDNS achieves $(2.44\%, 2.00\%)$ global performance improvement and $(11.20\%, 2.70\%)$ local performance improvement on the METR-LA dataset. Moreover, we observe STPGD-TDNS and STMIM-TDNS, two variants of our framework, respectively achieve the best attack performance on *PeMS-BAY* and *METR-LA* datasets, which further validate the superiority of our framework for flexibly integrate different adversarial perturbation methods. Overall, our adversarial attack framework successfully disrupts the traffic forecasting model to make biased predictions.

## 4.3 Ablation study

Then we conduct ablation study on our adversarial attack framework. Due to page limit, we report the result of STPGD-TDNS on the *PeMS-BAY* dataset. We consider two variants of our approach: (1) *w/o TDNS* that randomly choose victim nodes to attack, and (2) *w/o STPGD* that apply vanilla PGD noise to selected victim nodes. As reported in Table 2, we observe $(3.91\%, 6.28\%, 2.97\%, 3.60\%)$ and

Table 2: Ablation study on *PeMS-BAY*.

|  | G-MAE | L-MAE | G-RMSE | L-RMSE |
|---|---|---|---|---|
| non-attack | 1.975 | - | 4.0220 | - |
| w/o TDNS | 5.9024 | 4.8595 | 10.364 | 9.3635 |
| w/o STPGD | 4.5969 | 3.3876 | 8.4572 | 7.2949 |
| STPGD-TDNS | **6.1329** | **5.1647** | **10.6723** | **9.7003** |

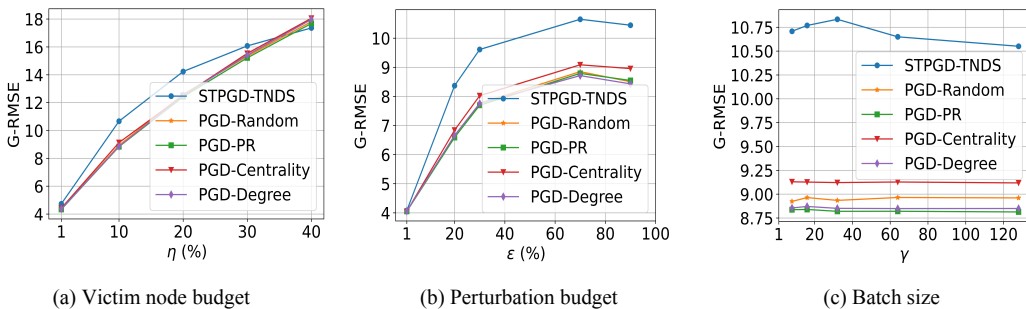

(a) Victim node budget      (b) Perturbation budget      (c) Batch size

Figure 2: Parameter sensitivity on *PeMS-BAY*.

$(33.41\%, 52.45\%, 26.19\%, 32.97\%)$ attack performance degradation on four metrics by removing our proposed TDNS and STPGD module, respectively. The above results demonstrate the effectiveness of the two-step framework. Moreover, we observe that the STPGD module plays a more important role in the adversarial spatiotemporal attack.

## 4.4 Parameter sensitivity

We further study the parameter sensitivity of the proposed framework, including the number of victim nodes $\eta$, the perturbation budget $\varepsilon$, and the batch size $\gamma$. Due to page limit, we report the result of G-RMSE on the *PeMS-BAY* dataset. We observe similar results by using other metrics and on the *METR-LA* dataset. Each time we vary a parameter, we set other parameters to their default values.

**Effect of $\eta$.** First, we vary the number of victim nodes from $0\%$ to $40\%$. As reported in Figure 2 (a), Our approach achieves the best attack performance with a limited victim node budget, and the advantage decrease when the attack can be applied to more nodes.

**Effect of $\varepsilon$.** Second, we vary the perturbation budget from $0\%$ to $90\%$. As shown in Figure 2 (b), the G-RMSE first increase and then slightly decrease. This is perhaps because the clip function in Equation 11 weakens the diversity of attack noises.

**Effect of $\gamma$.** Finally, we vary the batch size from $8$ to $128$, as illustrated in Figure 2 (c). We observe the adversarial attack is relatively stable to the batch size. However, too large batch size reduces the attack performance, which may induce over smooth of Equation 8.

## 4.5 Extended analysis under different attack settings

Table 3 reports the overall attack performance of our proposed approach against the original forecasting model and four PGD-based baselines under the white-box and black-box attack settings. For the white-box attack, since the attacker can fully access the data and model, we re-train the forecasting model without requiring estimating the latest traffic states. For the black-box attack, we adopt STAWNET [12] as the surrogate model. The experimental results are summarized in Table 3. First, we observe adversarial attacks significantly degrade the performance of the traffic forecasting model under both white-box and black-box settings. For examples, our approach achieves $((68.65\%, 66.12\%)$ and $(56.67\%, 50.78\%)$ global performance degradation compared with the vanilla forecasting model under white-box and black-box attack. Moreover, our approach consistently achieves the best attack performance against baselines. To be more specific, our approach yield $(4.61\%, 9.13\%)$ and $(1.70\%, 3.28\%)$ global performance improvement under the white-box setting and black-box setting, respectively. In addition, we observe higher attack effectiveness under

Table 3: Adversarial attack performance on *PeMS-BAY* under white-box and black-box settings.

| Metrics / Attack methods | White-box | | | | Black-box | | | |
|---|---|---|---|---|---|---|---|---|
| | G-MAE | L-MAE | G-RMSE | L-RMSE | G-MAE | L-MAE | G-RMSE | L-RMSE |
| non-attack | 2.0288 | - | 4.2476 | - | 1.9774 | - | 4.0219 | - |
| PGD-Random | 6.1477 | 5.0463 | 10.9217 | 9.5163 | 4.241 | 2.9738 | 7.3804 | 5.99 |
| PGD-PR | 6.1586 | 5.0713 | 10.7584 | 9.3405 | 4.4748 | 3.2605 | 7.9037 | 6.6306 |
| PGD-Centrality | 6.1723 | 5.0823 | 10.9468 | 9.5272 | 4.4859 | 3.3002 | 7.8795 | 6.6045 |
| PGD-Degree | 6.1507 | 5.0495 | 10.9375 | 9.5282 | 4.3577 | 3.1572 | 7.6159 | 6.2971 |
| **PGD-TDNS** | **6.4709** | **5.4953** | **12.1764** | **10.7262** | **4.5636** | **3.3543** | **8.1716** | **6.9388** |

Table 4: Performance of defense adversarial spatiotemporal attack on *PeMS-BAY*. ( Values in parentheses indicate std)

| Attack methods / Defense strategies | Non-attack | PGD-Random | PGD-PR | PGD-Centrality | PGD-Degree |
|---|---|---|---|---|---|
| Non-defense | **2.0288** | 6.1477 | 6.1586 | 6.1723 | 6.1507 |
| AT | 2.1156 | 2.5436 (0.0249) | 2.5539 (0.0375) | 2.5660 (0.0281) | 2.5394 (0.0279) |
| Mixup | 2.3090 | 2.7482 (0.0126) | 2.7573 (0.0241) | 2.7501 (0.0088) | 2.7788 (0.0234) |
| AT-TDNS | 2.0935 | **2.4695 (0.0036)** | **2.4463 (0.0075)** | **2.4549 (0.0023)** | **2.4474 (0.0069)** |

the white-box setting and lower attack effectiveness under the black-box setting compared to the grey-box setting. This makes sense as the white-box setting can fully access the data and label, while the black-box has more restrictive data accessibility and relies on the surrogate model to apply adversarial spatiotemporal attack.

## 4.6   Defense adversarial spatiotemporal attacks

Finally, we study the defense of adversarial spatiotemporal attacks. One primary goal of our study is to help improve the robustness of spatiotemporal forecasting models. Therefore, we propose to incorporate the adversarial training scheme for traffic forecasting models with our adversarial traffic states, denoted by *AT-TNDS*. We compare it with (1) conventional adversarial training (*AT*) [8] and (2) *Mixup* [24] with our adversarial traffic states. Note that we also tried other strategies, such as adding $L_2$ regularization, *etc.*, which fail to defend the adversarial spatiotemporal attack. The other state-of-the-art adversarial training methods, such as TRADE [25], cannot be directly applied in regression tasks. Please refer to Appendix E for more training details.

The results in G-MAE on the *PeMS-BAY* are reported in Table 4. Overall, we observe *AT* or *Mixup* can successfully resist the adversarial spatiotemporal attack, and *AT-TDNS* that combines the adversarial training scheme with our adversarial traffic states achieves the best defensive performance. The above results indicate the defensibility of adversarial spatiotemporal attacks, which should be further investigated to deliver a more reliable spatiotemporal forecasting service in the future.

## 5   Related work

**Spatiotemporal traffic forecasting**. In recent years, the deep learning based traffic forecasting model has been extensively studied due to its superiority in jointly modeling temporal and spatial dependencies [10, 11, 6, 26, 2, 12, 27, 28]. To name a few, STGCN [10] applied graph convolution and gated causal convolution to capture the spatiotemporal information in the traffic domain, ASTGCN [11] proposed a spatial-temporal attention network for capturing dynamic spatiotemporal correlations. As another example, GraphWaveNet [2] adaptively captures latent spatial dependency without requiring prior knowledge of the graph structure. The key objective of the above mentioned models is more accurate traffic forecasting. The vulnerability of spatiotemporal traffic forecasting models remains an under explored problem.

**Adversarial attack**. Deep neural networks have been proven vulnerable to adversarial examples [8, 14]. As an emerging direction, various adversarial attack strategies on graph-structured data have been proposed, including both target-attack and non-target attack [29, 30]. However, existing efforts on adversarial attacks mainly focus on classification tasks with static label [9, 24]. Only a few works study the vulnerability of GCN based spatiotemporal forecasting models under query-based attack [31] and generate adversarial examples based on evolutionary algorithms [32]. In this paper, we study the gradient based adversarial attack method against spatiotemporal traffic forecasting

models, which is model-agnostic and generalizable to various attack settings, *i.e.*, white-box attack, grey-box attack, and black-box attack.

# 6    Conclusion

This paper showed the vulnerability of spatiotemporal traffic forecasting models under adversarial attacks. We proposed a practical adversarial spatiotemporal attack framework, which is agnostic to forecasting model architectures and is generalizable to various attack settings. To be specific, we first constructed an iterative gradient guided node saliency method to identify a small time-dependent set of victim nodes. Then, we proposed a spatiotemporal gradient descent based scheme to generate real-valued adversarial traffic states by flexibly leveraging various adversarial perturbation methods. The theoretical analysis demonstrated the upper bound of the proposed two-step framework under human imperceptible victim node selection budget and perturbation budget constraints. Finally, extensive experimental results on real-world datasets verify the effectiveness of the proposed framework. The reported results will inspire further studies on the vulnerability of spatiotemporal forecasting models, as well as practical defending strategies for resisting adversarial attacks that can be deployed in real-world ITS systems.

## Acknowledgments and Disclosure of Funding

This work is supported by the National Natural Science Foundation of China under Grant No.62102110, and Foshan HKUST Projects (FSUST21-FYTRI01A, FSUST21-FYTRI02A).

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
