# A  Adversarial spatiotemporal attack under different settings

## A.1  Adversarial spatiotemporal attack under the white-box setting

Since the adversaries can fully access the data and label under the white-box setting, we directly use the real ground truth traffic states to generate the adversarial traffic states, as detailed in algorithm 2.

---

**Algorithm 2:** Adversarial spatiotemporal attack under the white-box setting

---

**Input:**  Previous traffic data, pre-trained spatiotemporal model $f_{\theta^*}(\cdot)$, pre-trained traffic state prediction model $g_\varphi(\cdot)$, maximum perturbation budget $\varepsilon$, victim node budget $\eta$, and iterations $K$.

**Result:**  Perturbed adversarial traffic states $\mathcal{H}'_{t-T+1:t}$.

/* Step 1:   Identify time-dependent victim nodes                    */

1  Compute the time-dependent node saliency $\mathcal{M}_t$ with $\mathcal{H}_{t-T+1:t}$ and $\mathbf{Y}_{t+1:t+\tau}$ by Equations 6-9;

2  Obtain the victim node set $S_t$ by Equation 10 ;

/* Step 2:   Attack with adversarial traffic state                   */

3  Initialize adversarial traffic state $\mathcal{H}'^{(0)}_{t-T+1:t} = \mathcal{H}_{t-T+1:t}$;

4  **for**  $i = 1$ *to* $K$ **do**

5  $\quad$ Generate perturbed adversarial features $\mathbf{X}'^{(i)}_{t-T+1:t}$ by Equation 11;

6  $\quad$ $\triangle\mathcal{H}'^{(i)}_{t-T+1:t} = ((\mathbf{X}'^{(i)}_{t-T+1:t} - \mathbf{X}^{(i)}_{t-T+1:t}) \cdot S_t, 0)$;

7  **end**

8  Return $\mathcal{H}'_{t-T+1:t} = \mathcal{H}_{t-T+1:t} + \triangle\mathcal{H}'_{t-T+1:t}$.

---

## A.2  Adversarial spatiotemporal attack under the black-box setting

The most restrictive black-box setting assumes limited accessibility to the target model and labels. Therefore, we first employ a surrogate model, which can be learned on the training data or query the traffic forecasting service [16, 17]. Then we generate adversarial traffic states based on the surrogate model to attack the target model. In details, we use a surrogate model to generate the adversarial traffic states based on algorithm 1, the generated adversarial traffic states can be used to attack the target model.

# B  Proof

In this section, we show the details of the proof. First, we recall the assumptions as follows: let the $k$-th layer embedding of spatiotemporal traffic models is

$$\mathbf{Z}_i^{(k+1)} = \sigma(\sum_{j \in \mathcal{N}_i} e_{ij}\mathbf{M}_j^{(k)}), \tag{13a}$$

where $\mathbf{Z}_i^{(k+1)}$ ($\mathbf{Z}^{(0)} = \mathcal{H}_{t-T+1:t}$) represents the embedding of node $v_i$ in $k+1$-th layer of the spatiotemporal forecasting model. $\mathbf{M}_i^{(k)} = \mathbf{Z}_i^{(k)}\mathbf{W}^{(k)}$, where $\mathbf{W}^{(k)}$ denotes the weight matrix for $k$-th layer of the forecasting model. $\sigma$ is an activation function, such as the sigmoid function, relu function, *etc.* $e_{ij}$ is the weight value used to aggregate node $j$'s neighbors. $\mathcal{N}_i$ represents the index used to keep track of node $j$'s neighbors. Let $\lambda$ denotes maximum weight bound in all layers of the forecasting model, where $\max_k \left\|\mathbf{W}^{(k)}\right\|_2 \leq \lambda, \forall k \in \{1, \cdots, L\}$. We denote that the maximum degree in graph $\mathcal{G}$ is $C$.

**Assumption 1** *The activation function $\sigma$ used in spatiotemporal traffic forecasting model is locally Lipschitz continuous as,*

$$\left\|\sigma(\sum_{j \in \mathcal{N}_i} e_{ij}\mathbf{M}_j^{(k)}) - \sigma(\sum_{j \in \mathcal{N}_i} e_{ij}\mathbf{M}'^{(k)}_j)\right\|_2 \leq \beta \left\|\sum_{j \in \mathcal{N}_i} e_{ij}\mathbf{M}_j^{(k)} - \sum_{j \in \mathcal{N}_i} e_{ij}\mathbf{M}'^{(k)}_j\right\|_p, \tag{14}$$

*where $\beta$ denotes parameter of the activation function in $f_\theta(\cdot)$.*

**Proof 1**

$$\left\| \mathbf{Z}^{(L)} - \mathbf{Z'}^{(L)} \right\|_2^2 = \sum_i \left\| \mathbf{Z}_i^{(L)} - \mathbf{Z'}_i^{(L)} \right\|_2^2$$

$$\leq \beta^2 \sum_i \left\| \sum_{j \in \mathcal{N}_i} e_{ij} \mathbf{M}_j^{(L-1)} - \sum_{j \in \mathcal{N}_i} e_{ij} \mathbf{M'}_j^{(L-1)} \right\|_2^2$$

$$\leq \beta^2 \sum_i |\mathcal{N}_i| \sum_{j \in \mathcal{N}_i} \left\| e_{ij}(\mathbf{M}_j^{(L-1)} - \mathbf{M'}_j^{(L-1)}) \right\|_2^2$$

$$\leq \beta^2 C \sum_i \sum_{j \in \mathcal{N}_i} \left\| \mathbf{M}_j^{(L-1)} - \mathbf{M'}_j^{(L-1)} \right\|_2^2$$

$$= \beta^2 C \sum_j |\mathcal{N}_j| \left\| \mathbf{M}_j^{(L-1)} - \mathbf{M'}_j^{(L-1)} \right\|_2^2$$

$$\leq \beta^2 C^2 \sum_j \left\| \mathbf{M}_j^{(L-1)} - \mathbf{M'}_j^{(L-1)} \right\|_2^2$$

$$\leq (\beta C \lambda)^2 \sum_j \left\| \mathbf{Z}_j^{(L-1)} - \mathbf{Z'}_j^{(L-1)} \right\|_2^2$$

$$\leq (\beta C \lambda)^{2L} \sum_i \left\| \mathbf{Z}_i^{(0)} - \mathbf{Z'}_i^{(0)} \right\|_2^2$$

$$= (\beta C \lambda)^{2L} \Big( \sum_{i \text{ is the victim node}} \left\| \boldsymbol{\mathcal{H}}_{(i),(t-T+1:t)} - \boldsymbol{\mathcal{H}'}_{(i),(t-T+1:t)} \right\|_2^2$$

$$+ \sum_{i \text{ is not the victim node}} \left\| \boldsymbol{\mathcal{H}}_{(i),(t-T+1:t)} - \boldsymbol{\mathcal{H}'}_{(i),(t-T+1:t)} \right\|_2^2 \Big)$$

$$\leq (\beta C \lambda)^{2L} \varepsilon^2 \eta$$

**Remarks**. Assumption 1 provides a more general activation function assumption. This assumption is met by the ReLU, sigmoid, tanh function [10, 11, 2] *etc.* We also noticed that [31] also analyzes traffic forecasting loss under query-based attack. Our theorem is different in that we first give the worst performance bound of an adversarial traffic forecasting attack, but [31] does not provide the worst performance bound. Second, our theorem is more general because we do not specify a specific activation function.

## C   Data statistics

We conclude the data statistics for two-real world datasets in Table 5.

Table 5: Data statistics

| Data | Sample | Nodes | Traffic events |
|------|--------|-------|----------------|
| *PeMS-BAY* | 34,272 | 325 | 16,937,700 |
| *METR-LA* | 52,116 | 207 | 7,094,304 |

## D  Evaluation metric

The Global MAE (G-MAE), Local MAE (L-MAE), Global RMSE (G-RMSE), Local RMSE (L-RMSE) are defined in Equations 16a-17b.

$$\text{G-MAE} = \frac{1}{m \times n} \sum_t \| f_\theta(\mathcal{H}'_{t-T+1:t}) - \mathbf{Y}_{t+1:t+\tau} \| \tag{16a}$$

$$\text{L-MAE} = \frac{1}{m \times n} \sum_t \| f_\theta(\mathcal{H}'_{t-T+1:t}) - f_\theta(\mathcal{H}_{t-T+1:t}) \|, \tag{16b}$$

$$\text{G-RMSE} = \sqrt{\frac{1}{m \times n} \sum_t \| f_\theta(\mathcal{H}'_{t-T+1:t}) - \mathbf{Y}_{t+1:t+\tau} \|^2} \tag{17a}$$

$$\text{L-RMSE} = \sqrt{\frac{1}{m \times n} \sum_t \| f_\theta(\mathcal{H}'_{t-T+1:t}) - f_\theta(\mathcal{H}_{t-T+1:t}) \|^2}, \tag{17b}$$

where $m$ represents the number of samples in test sets, and $n$ denotes the number of nodes.

## E  Defense adversarial traffic states

Given a spatiotemporal forecasting model $f_\theta(\cdot)$, the adversarial training in spatiotemporal traffic forecasting is defined as

$$\min_\theta \max_{\substack{\mathcal{H}'_{t-T+1:t} \\ t \in \mathcal{T}_{train}}} \sum_{t \in \mathcal{T}_{train}} \mathcal{L}(f_\theta(\mathcal{H}'_{t-T+1:t}), \mathbf{Y}_{t+1:t+\tau}), \tag{18}$$

where $\mathcal{L}(\cdot)$ is the loss function measuring the distance between the predicted traffic states and ground truth, and $\theta$ is parameters learned during the training stage. $\mathcal{T}_{train}$ denote the set of time steps of all training samples. We use strategies that include (1) adversarial training (AT) [8]. We use adversarial training with the PGD-Random adversarial attack method to generate the adversarial samples under white-box setting. (2) Mixup [24]. We randomly sample the clean and adversarial samples to train the forecasting model. The adversarial sample are also generated by PGD-Random method under white-box setting. (3). We use adversarial sampels generated by our method STPGD-TDNS under white-box setting to train the model.

## F  Further experiments

### F.1  Experiments on other models

The other spatiotemporal traffic forecasting models are summarized as follows. (1) STGCN [10] applies graph convolution and gated causal convolution to capture the spatiotemporal information in the traffic domain. (2) To overcome the spatiotemporal forecasting problem, ASTGCN [11] presented a spatial-temporal attention method for capturing dynamic spatiotemporal correlations. (3) MTGNN [33] created a self-learned node embedding for forecasting traffic conditions that is also not dependent on a pre-defined graph.

We report the evaluation results on other target models in Tables 6-8. By carefully selecting victim nodes, the attacker can achieve more effective attack performance with less attack budget. In particular, STPGD-TDNS achieves (62.23.80%, 55.86%) global performance improvement and (66.95.35%, 59.25%) local performance improvement on the PeMS-BAY dataset for MTGNN.

### F.2  Ablation study under white-box setting

Since selecting a few set as the victim nodes is important to attack traffic forecasting model, we conduct further ablation study to evaluate the method TDNS under the white-box setting. Table 9 reports the overall results on Gwnet under white-box attack.

Table 6: Grey-box attack on STGCN for *PeMS-BAY*

| Methods | G-MAE | L-MAE | G-RMSE | L-RMSE |
|---|---|---|---|---|
| non-attack | 2.8324 | - | 5.1708 | - |
| PGD-Random | 5.7924 | 4.0880 | 9.5659 | 8.0560 |
| PGD-PR | 9.6118 | 8.1697 | 15.4945 | 14.6314 |
| PGD-Centrality | 6.9712 | 5.1407 | 11.9507 | 10.7645 |
| PGD-Degree | 6.3903 | 4.3974 | 11.8196 | 10.6630 |
| MIM-Random | 6.0461 | 4.4043 | 9.8926 | 8.4604 |
| MIM-PR | 9.5573 | 8.1512 | 15.2504 | 14.3865 |
| MIM-Centrality | 6.9748 | 5.1906 | 11.6700 | 10.4777 |
| MIM-Degree | 6.5071 | 4.5425 | 11.8073 | 10.6640 |
| **STPGD-TDNS** | 9.3440 | 7.8039 | 5.1708 | 14.8150 |
| **STMIM-TDNS** | **10.2563** | **8.7318** | **5.1708** | **15.0358** |

Table 7: Grey-box attack on ASTGCN for *PeMS-BAY*

| Methods | G-MAE | L-MAE | G-RMSE | L-RMSE |
|---|---|---|---|---|
| non-attack | 2.3581 | - | 4.9165 | - |
| PGD-Random | 5.2302 | 3.1082 | 11.5757 | 10.4736 |
| PGD-PR | 5.2565 | 3.1282 | 11.6177 | 10.5154 |
| PGD-Centrality | 5.2260 | 3.1101 | 11.5842 | 10.4797 |
| PGD-Degree | 5.2504 | 3.1377 | 11.6332 | 10.5305 |
| MIM-Random | 5.1907 | 3.0609 | 11.4680 | 10.3509 |
| MIM-PR | 5.2080 | 3.0787 | 11.5024 | 10.3861 |
| MIM-Centrality | 5.1733 | 3.0569 | 11.4584 | 10.3409 |
| MIM-Degree | 5.2042 | 3.0900 | 11.5236 | 10.4065 |
| **STPGD-TDNS** | 5.2635 | 3.1476 | 11.6880 | 10.5896 |
| **STMIM-TDNS** | **5.2929** | **3.1799** | **11.7534** | **10.6579** |

Table 8: Grey-box attack on MTGNN for *PeMS-BAY*

| Methods | G-MAE | L-MAE | G-RMSE | L-RMSE |
|---|---|---|---|---|
| non-attack | 2.1501 | - | 4.2637 | - |
| PGD-Random | 5.4748 | 4.6839 | 9.5824 | 8.7328 |
| PGD-PR | 4.7997 | 3.8990 | 8.7011 | 7.7349 |
| PGD-Centrality | 5.6504 | 4.8921 | 9.6820 | 8.8529 |
| PGD-Degree | 4.9282 | 4.0396 | 8.8791 | 7.9403 |
| MIM-Random | 5.7671 | 4.9483 | 9.9446 | 9.1007 |
| MIM-PR | 4.8927 | 3.9385 | 8.9900 | 8.0265 |
| MIM-Centrality | 5.6832 | 4.8927 | 9.8080 | 8.9533 |
| MIM-Degree | 4.9599 | 4.0260 | 9.0387 | 8.0839 |
| **STPGD-TDNS** | 14.9606 | 14.8017 | 21.9354 | 21.7272 |
| **STMIM-TDNS** | **16.0254** | **15.9020** | **23.3589** | **23.1604** |

Table 9: Ablation study under white-box attack on Gwnet for *PeMS-BAY*

| Methods | G-MAE | L-MAE | G-RMSE | L-RMSE |
|---|---|---|---|---|
| non-attack | 2.0288 | - | 4.2476 | - |
| STPGD-Random | 6.1477 | 5.0463 | 10.9217 | 9.5163 |
| STPGD-PR | 6.1586 | 5.0713 | 10.7584 | 9.3405 |
| STPGD-Centrality | 6.1723 | 5.0823 | 10.9468 | 9.5272 |
| STPGD-Degree | 6.1507 | 5.0495 | 10.9375 | 9.5282 |
| STMIM-Random | 5.9524 | 4.8091 | 10.6488 | 9.1917 |
| STMIM-PR | 5.9311 | 4.7954 | 10.4354 | 8.9565 |
| STMIM-Centrality | 5.9159 | 4.7786 | 10.5948 | 9.1180 |
| STMIM-Degree | 5.9570 | 4.8085 | 10.6692 | 9.2136 |
| **STPGD-TDNS** | **6.4709** | **5.4953** | **12.1764** | **10.7262** |
| **STMIM-TDNS** | 6.3018 | 5.2733 | 11.8618 | 10.3729 |

## F.3 Experiments at different time intervals

We conduct further experiments at different time intervals, including 5 minutes, 10 minutes, 15 minutes, 30 minutes, and 45 minutes. We report the results at different time intervals compared with other baselines in Tables 11-15. Overall, as the time interval increases, the forecasting and adversarial attack performances decrease, as reported in Table 10.

For example, the G-MAE increases from 3.9458 to 6.1329 from a time interval of 5 minutes to a time interval of 60 minutes, with the attack performance degradation from 75.93% to 67.80%. One possible reason is that as the time interval increases, the forecasting error of the spatiotemporal model will increase. It is more challenging for the adversarial attack methods to estimate the target label to generate effective adversarial examples.

Table 10: Grey-box attack on Gwnet for *PeMS-BAY* at different minutes interval

|                        | 5 minutes | 10 minutes | 15 minutes | 30 minutes | 45 minutes | 60 minutes |
|------------------------|-----------|------------|------------|------------|------------|------------|
| non-attack             | 0.9496    | 1.1367     | 1.2747     | 1.6154     | 1.8872     | 1.9750     |
| STPGD-TDNS (ours)      | 3.9458    | 4.2924     | 3.6028     | 4.6629     | 5.2931     | 6.1329     |
| performance degradation| 75.93 %   | 73.46 %    | 64.62 %    | 65.36 %    | 64.34 %    | 67.80 %    |

Table 11: Grey-box attack on Gwnet for *PeMS-BAY* on 5 minutes interval

|                | G-MAE  | L-MAE  | G-RMSE  | L-RMSE  |
|----------------|--------|--------|---------|---------|
| non-attack     | 0.9496 |        | 1.7694  |         |
| PGD-Random     | 3.7926 | 3.0507 | 10.1258 | 9.9924  |
| PGD-PR         | 3.8226 | 3.0885 | 10.1880 | 10.0526 |
| PGD-Centrality | 3.7901 | 3.0586 | 10.1208 | 9.9950  |
| PGD-Degree     | 3.8302 | 3.0839 | 10.1733 | 10.0395 |
| **STPGD-TDNS** | **3.9458** | **3.2351** | **10.7429** | **10.6116** |

Table 12: Grey-box attack on Gwnet for *PeMS-BAY* on 10 minutes interval

|                | G-MAE  | L-MAE  | G-RMSE  | L-RMSE  |
|----------------|--------|--------|---------|---------|
| non-attack     | 1.1367 |        | 2.2430  |         |
| PGD-Random     | 4.2301 | 3.3311 | 10.9604 | 10.7417 |
| PGD-PR         | 4.2628 | 3.3769 | 11.0127 | 10.7993 |
| PGD-Centrality | 4.2234 | 3.3378 | 10.9677 | 10.7543 |
| PGD-Degree     | 4.2779 | 3.3778 | 11.0364 | 10.8219 |
| **STPGD-TDNS** | **4.2924** | **3.4586** | **11.4178** | **11.2231** |

Table 13: Grey-box attack on Gwnet for *PeMS-BAY* on 15 minutes interval

|  | G-MAE | L-MAE | G-RMSE | L-RMSE |
|---|---|---|---|---|
| non-attack | 1.2747 |  | 2.5761 |  |
| PGD-Random | 3.6073 | 2.7355 | 9.0194 | 8.6871 |
| PGD-PR | 3.6011 | 2.7540 | 8.9609 | 8.6240 |
| PGD-Centrality | 3.6004 | 2.7314 | 9.0132 | 8.6853 |
| PGD-Degree | **3.6206** | 2.7510 | 8.9892 | 8.6531 |
| **STPGD-TDNS** | 3.6028 | **2.7798** | **9.1164** | **8.7607** |

Table 14: Grey-box attack on Gwnet for *PeMS-BAY* on 30 minutes interval

|  | G-MAE | L-MAE | G-RMSE | L-RMSE |
|---|---|---|---|---|
| non-attack | 1.6154 |  | 3.2933 |  |
| PGD-Random | 3.4294 | 2.3903 | 6.9265 | 6.0358 |
| PGD-PR | 3.4214 | 2.3999 | 6.8360 | 5.9331 |
| PGD-Centrality | 3.4666 | 2.4328 | 7.0459 | 6.1729 |
| PGD-Degree | 3.4190 | 2.3731 | 6.8721 | 5.9837 |
| **STPGD-TDNS** | **4.6629** | **3.7733** | **8.9025** | **8.1430** |

Table 15: Grey-box attack on Gwnet for *PeMS-BAY* on 45 minutes interval

|  | G-MAE | L-MAE | G-RMSE | L-RMSE |
|---|---|---|---|---|
| non-attack | 1.8872 |  | 3.8593 |  |
| PGD-Random | 3.6825 | 2.4705 | 7.3557 | 6.2334 |
| PGD-PR | 3.6789 | 2.4925 | 7.3180 | 6.1898 |
| PGD-Centrality | 3.6872 | 2.4897 | 7.4748 | 6.3791 |
| PGD-Degree | 3.7254 | 2.5300 | 7.4270 | 6.3325 |
| **STPGD-TDNS** | **5.2931** | **4.3660** | **9.7466** | **8.9135** |