# OpenReview forum: "Practical Adversarial Attacks on Spatiotemporal Traffic Forecasting Models"
_NeurIPS.cc/2022/Conference — NeurIPS 2022 Accept_

### Official Review · Reviewer_EYep · 2022-06-27

**Rating:** 7
**Confidence:** 4
**Soundness:** 3 good
**Presentation:** 3 good
**Contribution:** 3 good

**Summary:**

The paper tackles the problem of adversarial attacks against “traffic forecasting models”, i.e., models that must predict the conditions of “physical traffic” (e.g., cars) in real world settings. The main contribution is a generic framework of adversarial attacks against such systems, which can be declined to cover both white- and black-box adversarial settings. A comprehensive experimental evaluation on real world data validates the proposal. The paper is also enriched by an ablation study, and by strong theoretical analyses that support the overall findings

**Questions:**

I thank the authors for their paper, of which I particularly appreciated the research direction: instead of focusing on the (overused) attacks against image classifiers, the paper considers a completely different (but still realistic) setting. I have some comments that the authors can address in a rebuttal, which are reported below.

*Unclear definitions of White/Black-box attacks.* In Section 3.2 the authors describe the potential “variations” of the proposed attack by using the well-known “white/black-box” terminology. However, it is not clear what are the actual assumptions of the considered attacker. For example, the white-box scenario is described as follows: “White-box attack. Since the adversaries can fully access the data and labels under the white-box setting, we directly use the real ground truth traffic states to guide the generation of adversarial traffic states.”. Does this mean that the “white-box” attack assumes an attacker with access to the training data and all labels? Or does this also imply that the attacker knows everything about the target model (i.e., learned parameters, weights and architecture)? Similarly, for black-box attacks the paper states that: “The most restrictive black-box setting assumes limited accessibility to the target model and labels. Therefore, we first employ a surrogate model, which can be learned from the training data or by querying the traffic forecasting service [15, 16]. Then we generate adversarial traffic states based on the surrogate model to attack the targeted traffic forecasting model.” The authors should elucidate if they assume an attacker who can “query” the target model (and whether such queries are constrained or not) or who has no access whatsoever to the targeted model, but only to a subset of the training data (in which case, it is more like a “no-box” attack [A]). For completeness, I have also checked the appendix and there is no mentioning of such details.

*Realistic Feasibility.* I would be delighted if the authors could integrate (perhaps in an appendix) an analysis of the realistic feasibility of the proposed attacks. For example, consider the following statement in the introduction: “How to identify the subset of salient victim nodes with a limited attack budget to maximize the attack effect is the first challenge.” Why would a realistic attacker do that? If, as shown in Figure 1, just by injecting perturbations on “randomly selected” nodes results in a successful attack, then why going further? Would an attacker really opt for such a strategy? What is the cost and potential benefit of such a strategy?
Indeed, I stress that real attacker operate with a cost/benefit mindset. Considered that the paper tackles a new application domain, it would be impactful to show that some attacks may require little preparation, but can lead to significant performance degradation. Of course, it is still valuable to analyze “worst-case” scenarios, but it’s important to differentiate from attacks that are more likely to occur (e.g., because they are cheaper to stage) from those who are less likely to appear (e.g., because they require a huge resource investment, which can potentially be superior than what the attacker stands to gain).

Some additional issues:

•	The following statement in the Introduction requires additional back-up: “Machine learned spatiotemporal forecasting models have been widely adopted in modern Intelligent Transportation Systems (ITS) to provide accurate and timely prediction of traffic dynamics, e.g., traffic flow [1], traffic speed [2], and the estimated time of arrival [3].” The problem is that [1,2,3] are research papers, and cannot be used to substantiate the claim that ML-related proposals are “widely adopted in modern ITS”. At best, they are well-studied in research.

•	*Training/testing time?* The paper reports that “All experiments are implemented with PyTorch and performed on a Linux server with 4 RTX 3090 GPUs.”. I am genuinely curious of how long it took to train the corresponding models on such hardware. Were the GPUs used in parallel, or did the experiments consider a single GPU “per run”?

EXTERNAL REFERENCES

[A]: Li, Qizhang, Yiwen Guo, and Hao Chen. "Practical no-box adversarial attacks against dnns." Advances in Neural Information Processing Systems 33 (2020): 12849-12860.


**Limitations:**

I did not see any limitation mentioned in the main paper, but I **do** have one comment on this matter: the suitability of the considered datasets. I acknowledge that finding proper data is difficult in this domain, and I also acknowledge that the theoretical arguments are well-founded. However, the two considered datasets are either 5 or 10 years old. Such "old-age" can (slightly) impair the real world implications of the paper findings.

**Strengths And Weaknesses:**

**Originality:** high. I agree that there is limited work done in this specific domain, and the strong analysis and comprehensive evaluation are novel contributions.

**Quality:** high. Despite lacking in some minor details, the arguments are well-supported.

**Clarity:** average. The English text is good enough to allow a reader to understand the paper, but not exceptional. Presentation-wise, the paper is also appreciable.

**Significance:** high. Albeit the findings are not counterintuitive (ultimately, the paper shows that “yet another application of ML can be thwarted via adversarial examples”), the considerations of a under-investigated deployment scenario of ML is commendable, and future work can greatly benefit from this paper.

---

> ### Author Response · Authors · 2022-08-02
> **Response to Reviewer EYep**
>
> We highly appreciate your high-quality review and valuable suggestions. We are pleased that the reviewer recognized our contributions to the adversarial attack on the traffic forecasting domain.
>
>
>
> ***
> > [Q1] “Unclear definitions of White/Black-box attacks”
>
> [Response]
> We have presented the detailed definition of white-box, grey-box, and Black-box attacks in section 2.2.
> “Based on the amount of information the attacker can access, the adversarial attack can
> be categorized into three classes. (1) White-box attack. The attacker has full access to the system, including parameters and gradients of the target model, the input data, and the label. (2) Grey-box attack. The attacker can partially access the system, including the target model and training input data, except the labels. (3) Black-box attack. The attacker can only access training input data but cannot access the target model and labels.”
> ***
>
> > [Q2] ”Realistic Feasibility.”
>
> [Response]
> In real-world traffic systems, the traffic data is generated from geo-distributed data sources (e.g., sensors). Perturbing different geo-located data requires hacking different sensors, which may be expensive. Therefore, we consider the attack budget as a critical constraint in adversarial attacks on traffic forecasting models. For the attack strategy, our method achieves (15.80%, 15.39%) global performance improvement and (23.35%, 17.19%) local performance improvement on the PeMS-BAY dataset compared to the random attack baseline. By carefully selecting victim nodes, the attacker can achieve more effective attack performance with less attack budget. We have clarified the above concerns in the Appendix F .1.
>
> ***
> > [Q3] “The following statement in the Introduction requires additional back-up: “Machine learned spatiotemporal forecasting models have been widely adopted in modern Intelligent Transportation Systems (ITS) to provide accurate and timely prediction of traffic dynamics, e.g., traffic flow [1], traffic speed [2], and the estimated time of arrival [3].” The problem is that [1,2,3] are research papers, and cannot be used to substantiate the claim that ML-related proposals are “widely adopted in modern ITS”. At best, they are well-studied in research.”
>
> [Response]
> Thanks for your suggestions. We introduced more spatiotemporal forecasting models deployed in the industry, including Google Maps [2] and Baidu Maps [1] in Section 1. As reported by the company development team, the described models have been deployed in the production environment. Besides, Google Maps also applied the graph-based model to predict the estimated arrival times (DeepMind Blog: https://www.deepmind.com/blog/traffic-prediction-with-advanced-graph-neural-networks).
>
>
> ***
> > [Q4] ”Training/testing time? The paper reports that “All experiments are implemented with PyTorch and performed on a Linux server with 4 RTX 3090 GPUs.”. I am genuinely curious of how long it took to train the corresponding models on such hardware. Were the GPUs used in parallel, or did the experiments consider a single GPU “per run”?”
>
> [Response]
> For the training, each epoch takes about 27 minutes. For testing, it will take about 5 minutes to test an attack method. The experiments  are conducted on a single GPU.
> ***
>
> References:
>
> [1] Liao, Binbing, et al. "Deep sequence learning with auxiliary information for traffic prediction." Proceedings of the 24th ACM SIGKDD International Conference on Knowledge Discovery & Data Mining. 2018.
>
> [2] Derrow-Pinion, Austin, et al. "Eta prediction with graph neural networks in google maps." Proceedings of the 30th ACM International Conference on Information & Knowledge Management. 2021.
>
> ***

---

> > ### Comment · Reviewer_EYep · 2022-08-03
> > **Still not convinced about the threat model**
> >
> > The rebuttal has addressed some of my concerns, but the most important one still remains.
> >
> > Specifically, I invite the authors to clearly state what they imply by "full access to the system" and, specifically, "the input data".
> >
> > This is because "full access" implies both "read and write" access -- and having "write access" also to the "label" means that an attacker can technically just change the output of the ML model after a prediction has been made. In other words: an attacker with such a 'power' would launch a different a much more dangerous attack than the one described in the paper. Finally, what is the "system"? Is it just the ML model, or the entire pipeline?
> >
> > With regards to "input data", do the authors imply the training dataset or the input data sent after the model has been deployed?
> >
> > These details are crucial to gauge the realistic value of the envisioned scenario, and I endorse the authors to be extremely precise in defining the considered threat model. In fact, I strongly oppose using the "white/black-box" terminology to define a threat model, as such terms only focus on the knowledge of the attacker and can be misleading.

---

> > > ### Author Response · Authors · 2022-08-05
> > > **Response to comments about the threat model**
> > >
> > > The full access to the system refers to the attackers can read the model including the whole model architecture, model parameters, gradients, model outputs, and the attackers can access the traffic forecasting input data as well as the corresponding labels in the testing stage. We agree with the reviewer that an attacker with read and write access can apply more dangerous attacks. We have revised the description of accessibility in the manuscript from Line 88 to 96.
> > >
> > > The system refers to the entire ML pipeline of the traffic forecasting model after deployed, including the geo-distributed traffic data, the feature constructor, and the well trained forecasting model.
> > >
> > > The input data refers to the data collected from geo-distributed sensors in the testing stage. For a time step t, the input data including the traffic states of all sensors from t-T+1 to t-1 time steps.
> > >
> > > For the threat model, we have added a more precise definition in Section 2.2 and 2.3.
> > >
> > > _Three types attack._
> > > “
> > > Note the adversarial attack happened in the testing stage, and the attackers cannot manipulate the forecasting model or its output. On the benign testing set, the forecasting model can perform well.  Based on the amount of information the attacker can access in the testing stage, the adversarial attack can be categorized into three classes.
> > >
> > > White-box attack. The attacker can fully access the target model, including the model architecture, the model parameters, gradients, model outputs, the input traffic states, and the corresponding labels.
> > >
> > >  Grey-box attack. The attacker can partially access the system, including the target model and the input traffic states, but without the labels.
> > >
> > > Black-box attack. The attacker can only access the input traffic states,  query the outputs of the target model or leverage a surrogate model to craft the adversarial examples.
> > > “
> > >
> > >
> > > _Attack goal._
> > > ” The attacker aims to craft adversarial traffic states to fool the spatiotemporal forecasting model to derive biased predictions. ” Formally, given a spatiotemporal forecasting model $f_\theta(\cdot)$, the adversarial attack against spatiotemporal traffic forecasting is defined as in Equation 4.  More detailed definition of the attack goal, please refer to line 106-111

---

> > > > ### Comment · Reviewer_EYep · 2022-08-05
> > > > **Acknowledgement**
> > > >
> > > > I thank the authors for their clarification. My overall score has not changed (it is still a 7), but I am more confident of the paper's contribution so I will take their response into account for any future discussion.

---

### Official Review · Reviewer_rQBx · 2022-07-10

**Rating:** 6
**Confidence:** 2
**Soundness:** 3 good
**Presentation:** 3 good
**Contribution:** 3 good

**Summary:**

This paper explores the vulnerability of spatiotemporal traffic forecasting models. They proposed a practical adversarial spatiotemporal attack framework. Experiments are conducted to verify the effectiveness.

**Questions:**

please see the weakness

**Ethics Review Area:**

["Privacy and Security (e.g., consent)"]

**Limitations:**

please see the weakness

**Strengths And Weaknesses:**


Strengths:
1. The first attemp to attack spatiotemporal traffic forecasting models.
2. The code is released, and thus readers can reproduce the results.
3. The demonstration is given from empirical and theoretical view.

Weakness:
1. When the attacks are extended to black-box setting, they  employ a surrogate model, which is trained via querying the threat models. This will lead to a large number of queris, which reduces the efficiency. The authors should discuss this point.

---

> ### Author Response · Authors · 2022-08-02
> **Response to Reviewer rQBx**
>
> We highly appreciate your positive opinions on the methodology and the insightful comments.
>
>
>
> ***
> > [W1] ”When the attacks are extended to black-box setting, they employ a surrogate model, which is trained via querying the threat models. This will lead to a large number of queris, which reduces the efficiency. The authors should discuss this point.”
>
> [Response]
> Thanks for the insightful comment. In practice, the gradient of the models might be estimated using a surrogate model (transfer-based attack) [1] or zeroth-order optimization approaches (query-based attack) [2]. It is true that it may require a massive number of queries in the query-based attack.  However, in the training stage, we train the surrogate model in a  similar way with the threat model without querying the threat models in the transfer-based attack setting. Furthermore, in the inference stage, we use a surrogate model to estimate the gradient of the threat models to generate the adversarial examples. We left the query-based attack overhead under the black-box setting as future work.
> ***
>
> References:
>
> [1] 	Yanpei Liu, et al. “Delving into Transferable Adversarial Examples and Black-box Attacks”. In International Conference on Learning Representations. 2017.
>
> [2] Cheng, Shuyu, et al. "Improving black-box adversarial attacks with a transfer-based prior." Advances in neural information processing systems. 2019.
> ***

---

> ### Author Response · Authors · 2022-08-08
> **Response to  Reviewer rQBx**
>
> Dear reviewer rQBx:
>
> We thank you for the precious review time and valuable comments. We have provided corresponding responses and results, which we believe have covered your concerns. We hope to further discuss with you whether or not your concerns have been addressed. Please let us know if you still have any unclear parts of our work.
>
> Best,
>
> NeurIPS 2022 Conference Paper312 Authors

---

### Official Review · Reviewer_3JHx · 2022-07-10

**Rating:** 4
**Confidence:** 3
**Soundness:** 2 fair
**Presentation:** 3 good
**Contribution:** 2 fair

**Summary:**

The paper proposes a gradient-based adversarial attack method on traffic forecasting models. They propose two-stage frameworks for practical adversarial spatiotemporal attacks, demonstrating the performance degradation.  The proposed method can selectively attack the most sensitive nodes in the graph, based on the magnitude of the gradient to generate more effective attack results.

**Questions:**

- Why do we only keep the negative saliency score (line 143)? What is different if we take the absolute values?

- Why do we leave the topology of G_{t} immutable, when its adjacency relationship can be learned by the model parameter, which is mutable? (line 102)

- I am wondering if there is any difference in performance depending on the time interval (e.g., 5 minutes, 10 minutes, 1 hour, … ), can you explain more?


**Limitations:**

- There is no significant difference between AT and AT-TDNS, which is less than 0.03 G-MAE score. Multiple runs to produce mean+-std should be examined. (Table 4)

- Minor suggestion: Legends of Figure 1 b-c should be brought outside, as they interfere with the contents.

- I don’t think the white-box attack is realistic, since the target model is a time-series model that predicts the upcoming states.

- The authors didn’t explicitly address the limitations of the work. As said previously, the authors could include connectivity-based adversarial samples to improve the work. Also, the authors are encouraged to propose more novel ideas for acceptance for NeurIPS.


**Strengths And Weaknesses:**

Strength"
- The problem is well motivated and described, and the papers are well written.
-  Extensive experiments are conducted to demonstrate the effectiveness of an attack.



Weaknesses:

- This paper lacks originality in that it proposes no novel or new concepts but just simply modify the existing adversarial attack methods little to deal with spatiotemporal forecasting models. The only novel part of the paper is the newly proposed method of TDNS, but it is still a mere extension of PGD algorithm and taking top k nodes as the victim nodes. Other than the lack of novelty, the paper was nicely written and easy to follow. I personally do not think the methods proposed in this paper are either groundbreaking or significant for future research.

- I find there is no special attack designed for the time-series graph data. If we consider every time step as an observation, the attack design is very similar to PGD. And selecting the most sensitive node is similar to other data types.

- The proposed method is only applicable for traffic forecasting models; And, I cannot find the explicit expression of L_{adv}.

- The deviation of \phi from the pre-trained forecasting model was not well described. Suppose \phi is exactly the same with \theta^{\asterisk}. In that case, the derivative of the loss w.r.t its input is just the derivative of random noise (\delta) at the output, making the prediction far away from its first prediction after some iterations, which is not an effective attack approach.

- There is no exploration of the edge perturbation, or at least compared with the method that perturbation the edge of graph data.

- Estimating the effectiveness of the attack method on more target models should be brought to the main table to show its generality.

- I wonder why they kept the topology (adjacency matrix, connectivity of the road network) immutable, when perturbing the test data to generate adversarial samples. I acknowledge the point made in the paper that since a graph-based network diffuses the node features across the network, making perturbations to a subset of nodes will be sufficient for successful adversarial attack. As far as I know, graph construction is a major part of the GNN-based spatiotemporal forecasting. So the authors should include the generation of adversarial samples perturbing the original adjacency relation, at least provide an acceptable reason for dropping out that approach.

---

> ### Author Response · Authors · 2022-08-02
> **Response to Reviewer 3JHx Part 3  [3/ 3]**
>
> ***
> > [Q3] “I am wondering if there is any difference in performance depending on the time interval (e.g., 5 minutes, 10 minutes, 1 hour, … ), can you explain more?”
>
> [Response] Thanks for the insightful question. We have conducted more experiments at different time intervals, ranging from 5 minutes to 60 minutes. The performance of the traffic forecasting model under different time intervals in terms of G-MAE is reported below. Overall, as the time interval increases, the forecasting and adversarial attack performances decrease.
>
> For example, the G-MAE increases from 3.9458 to 6.1329 from a time interval of 5 minutes to a time interval of 60 minutes, with the attack performance degradation from 75.93% to 67.80%.  One possible reason is that as the time interval increases, the forecasting error of the spatiotemporal model will increase. It is more challenging for the adversarial attack methods to estimate the target label to generate effective adversarial examples.
>
>
> |    | 5 minutes | 10 minutes   | 15 minutes  | 30 minutes     | 45 minutes     |60 minutes     |
> | :----:       |    :----:   |          :----: | :----:   |          :----: | :----: | :----: |
> | non-attack      | 0.9496       | 1.1367   |   1.2747    | 1.6154  | 1.8872 | 1.9750|
> | STPGD-TDNS （ours）   |   3.9458   |  4.2924   |  3.6028   |  4.6629  |  5.2931|  6.1329|
> | performance degradation   |   75.93 %   |  73.46 %  | 64.62 %  |  65.36 %  | 64.34 %  | 67.80 %   |
>
> Moreover, the experiments result under 5 minutes compared to other baselines is shown as follows. The results on more time interval is reported in Appendix F.
>
> | Attack methods     | G-MAE | L-MAE     | G-RMSE | L-RMSE     |
> | :----:       |    :----:   |          :----: | :----:   |          :----: |
> | non-attack      | 0.9496       | -   |   1.7694     | -  |
> | PGD-Random   |   3.7926  |  3.0507  |   10.1258 |  9.9924    |
> | PGD-PR  |   3.8226  |  3.0885   | 10.1880  | 10.0526 |
> | PGD-Centrality   |  3.7901  |  3.0586    |  10.1208    | 9.9950 |
> | PGD-Degree   |   3.8302   | 3.0839    |    10.1733   |   10.0395 |
> | STPGD-TDNS   |   3.9458   | 3.2351 |  10.7429   |  10.6116 |
>
> ***
>
> > [L1]“There is no significant difference between AT and AT-TDNS, which is less than 0.03 G-MAE score. Multiple runs to produce mean+-std should be examined. (Table 4)”
>
> [Response] In traditional adversarial training (AT), all the features (nodes in our venues) would be added with the adversarial perturbations. However, traditional adversarial training (adding all features with adversarial perturbations) this would lead to severe overfitting problems. To solve this problem, we only chose a few nodes as the target nodes (adding adversarial perturbations) by our method TNDS. The experiments in section 4.6 demonstrate that using our strategy to choose nodes can achieve the best performance compared to randomly selecting nodes.
>
> According to the reviewer's suggestion, we have re-run the experiment 10 times and calculated the mean and std, as reported below. The results demonstrate AT-TDNS is more stable.
>
>
> | Attack methods     | Non-attack | PGD-Random     | PGD-PR | PGD-Centrality     | PGD-Degree|
> | :----:       |    :----:   |          :----: | :----:   |          :----: | :----: |
> | Non-defense     |  2.0288      | 6.1477   |   6.1586      | 6.1723   |6.1507 |
> | AT   |  2.1156  |  2.5436 (0.0249)  |   2.5539 (0.0375) |  2.5660 (0.0281)    | 2.5394 (0.0279)|
> | Mixup  |   2.3090  |  2.7482 (0.0126)   | 2.7573 (0.0241)  | 2.7501 (0.0088) | 2.7788 (0.0234)|
> | AT-TDNS   |  2.0935  |  2.4695 (0.0036)    |  2.4463 (0.0075)    | 2.4549 (0.0023) | 2.4474 (0.0069) |
>
>
> ***
> > [L2] ”Minor suggestion: Legends of Figure 1 b-c should be brought outside, as they interfere with the contents.”
>
> [Response]
> Thanks for the suggestion. We have reorganized the image in the manuscript accordingly.
> ***
> >[L3] ” I don’t think the white-box attack is realistic, since the target model is a time-series model that predicts the upcoming states”
>
> [Response]
> We strongly agree with the reviewer! Since the ground truth (i.e., future traffic states) under the spatiotemporal traffic forecasting setting is unavailable at run-time, the practical adversarial spatiotemporal attack primarily falls into the grey-box attack setting. However, little research has discussed the adversarial attack on traffic forecasting models. To this end, we began with the white-box attack setting and discussed how to apply adversarial attacks under the grey-box and black-box settings in Section 3.2. Moreover, investigating white-box attacks is still beneficial to help us understand how adversarial attack works and can help improve the robustness of spatiotemporal traffic forecasting models (e.g., applying adversarial training).
>
>
>
>
> ***

---

> ### Author Response · Authors · 2022-08-02
> **Response to Reviewer 3JHx Part 2 [2/3]**
>
> ***
>
>
> > [W3.2] "I cannot find the explicit expression of L_{adv}."
>
> [Response] Thanks for your detailed comments to help us improve the quality of the paper. The L_{adv} is calculated similarly to the loss function L (such as MAE and RMSE ). Specifically, the input will be updated by gradient-based attack methods [7] after multi-step updates. The final adversarial loss  is the  distance between the predictions of adversarial examples $f_{\theta }(\mathbfcal{\tilde{H}}\_{t-T+1:t})$ and the target $\mathbf{\tilde{Y}}\_{t+1:t+\tau}$.  Besides, we found that the subscription ‘adv’, which is used to emphasize that this is the adversarial loss, is indeed not coordinated with $\mathcal{L}$ in the previous sections, so we have updated the paper accordingly by changing $\mathcal{L}_{adv}$ to $\mathcal{L}$. The explicit expression is now discussed from line 133 to line 140.
>
> ***
>
> > [W4] “The deviation of \phi from the pre-trained forecasting model was not well described. Suppose \phi is exactly the same with \theta^{\asterisk}. In that case, the derivative of the loss w.r.t its input is just the derivative of random noise (\delta) at the output, making the prediction far away from its first prediction after some iterations, which is not an effective attack approach.”
>
> [Response] Thanks for your detailed question. The clean examples will be added with adversarial perturbation with multi-steps. In addition, random noise (\delta) is used to increase the diversity of the attack direction. So its input is always the derivative of adversarial examples at the output. Besides, the explicit process is discussed from line 139 to line 143.
>
>
> ***
>
> > [W5] & [W7] & [Q2] & [L4] “There is no exploration of the edge perturbation, or at least compared with the method that perturbation the edge of graph data.”, “ I wonder why they kept the topology (adjacency matrix, connectivity of the road network) immutable,” and   “Why do we leave the topology of G_{t} immutable, when its adjacency relationship can be learned by the model parameter, which is mutable? (line 102)
> ”
>
> [Response] Thanks for your insightful questions. The reason we don’t consider edge perturbation in this work is two-fold.
>
> First, the connectivity of the traffic network in the physical world is usually considered static, while the time-varying traffic dynamic features are collected from geo-distributed sensors. For generating adversarial examples, we argue modifying the traffic network topology in the physical world is difficult and easy to be detected. It is much more practical and meaningful to perturb time-dependent node features.
>
> Second, for state-of-the-art spatiotemporal forecasting models such as Graph Wave Net [9], the adjacency matrix of the graph is regarded as a part of the parameter and learned by the model in an end-to-end way [9]. In such a scenario, based on the definition of adversarial attack, the graph topology is a part of the model parameter and is fixed in the inference stage [10]. The attackers cannot perturb the model parameter but craft some adversarial examples to fool the model [8].
>
>
>
>
>
> ***
> > [W6] “Estimating the effectiveness of the attack method on more target models should be brought to the main table to show its generality.”
>
> Thanks for your suggestion. Due to the page limit, we have reported the experimental results on other target models in Appendix F.1. We have added an explanation in line 514 to line 517.
> ***
>
> > [Q1] “Why do we only keep the negative saliency score (line 143)? What is different if we take the absolute values?”
>
> [Response]
> The saliency score in line 143 is defined as non-negative, where larger score indicates more salient node.
> ***
> References:
>
> [7] Madry, Aleksander, et al. "Towards Deep Learning Models Resistant to Adversarial Attacks." International Conference on Learning Representations. 2018.
>
> [8] Xu, Han, et al. "Adversarial attacks and defenses in images, graphs and text: A review." International Journal of Automation and Computing. 2020.
>
> [9] Zonghan Wu, et al. “Graph WaveNet for Deep Spatial-Temporal Graph Modeling. “ International Joint Conference on Artificial Intelligence. 2019.
>
>
> [10] Yu, Bing et al. "Spatio-temporal graph convolutional networks: a deep learning framework for traffic forecasting." Proceedings of the 27th International Joint Conference on Artificial Intelligence. 2018.

---

> ### Author Response · Authors · 2022-08-02
> **Response to Reviewer 3JHx Part 1 [ 1/ 3]**
>
> We appreciate the reviewer recognized that our problem is well-motivated and the paper is well-written. We also thank the reviewer’s detailed questions. Please find a point-to-point response to the reviewer’s comments below.
>
>
> ***
> > [W1] ”This paper lacks originality in that it proposes no novel or new concepts but just simply modify the existing adversarial attack methods little to deal with spatiotemporal forecasting models. The only novel part of the paper is the newly proposed method of TDNS, but it is still a mere extension of PGD algorithm and taking top k nodes as the victim nodes. Other than the lack of novelty, the paper was nicely written and easy to follow. I personally do not think the methods proposed in this paper are either groundbreaking or significant for future research.
> ”
>
> [Response] Spatiotemporal traffic forecasting models have become a cornerstone of Intelligent Transportation Systems (ITS) and modern online maps, e.g., Google Maps [1] and Baidu Maps [2]. However, existing methods assume a reliable and unbiased forecasting environment. As also supported by Reviewer EYep, in this paper, we considered a completely different but still realistic setting compared with adversarial attacks, and proposed a generic framework of adversarial attacks against such spatiotemporal systems. Empirically analysis successfully proved the vulnerability of existing traffic forecasting models.
>
> More in detail, the proposed framework is not a simple extension of PGD, but a generic framework that other gradient-based methods such as MIM [3] , DIM [4] can also be integrated. In this paper, we use PGD to demonstrate the effectiveness of our framework. Furthermore, how identifying the time-dependent Top-k victim node set is a non-trivial problem and rarely considered by previous models. In this paper, we proposed an iterative gradient-guided node saliency method that incorporates both spatial and temporal information to identify the time-dependent set of victim nodes dynamically.
> ***
> > [W2] "I find there is no special attack designed for the time-series graph data. If we consider every time step as an observation, the attack design is very similar to PGD. And selecting the most sensitive node is similar to other data types.”
>
> [Response]
> We agree with the reviewer that one major drawback of most adversarial attack methods such as PGD, MIM, and DIM are static. In this work, we introduced the iterative gradient-guided node saliency method to identify the set of victim nodes in a time-dependent way, which implicitly incorporates the temporal dynamics in networked traffic data.
>
> Moreover, the adversarial attack tasks on other data types such as images are mostly classification tasks, where choosing the sensitive pixel has a clear classification function and target. However, traffic forecasting is naturally a regression problem, which requires a newly designed function (adversarial loss function) and target (adversarial nodes ).
> ***
> > [W3.1] "The proposed method is only applicable for traffic forecasting models."
>
> [Response]
> Our framework can be generalized to attack other spatiotemporal tasks, such as air quality prediction [5] and weather prediction [6]. Please refer to the response to Reviewer 7aqp [W1].
>
> ***
> References:
>
> [1] Derrow-Pinion, Austin, et al. "Eta prediction with graph neural networks in google maps." Proceedings of the 30th ACM International Conference on Information & Knowledge Management. 2021.
>
> [2] Liao, Binbing, et al. "Deep sequence learning with auxiliary information for traffic prediction." Proceedings of the 24th ACM SIGKDD International Conference on Knowledge Discovery & Data Mining. 2018.
>
>
>
> [3] Dong, Yinpeng, et al. "Boosting adversarial attacks with momentum." Proceedings of the IEEE conference on computer vision and pattern recognition. 2018.
>
> [4] Xie, Cihang, et al. "Improving transferability of adversarial examples with input diversity." Proceedings of the IEEE/CVF Conference on Computer Vision and Pattern Recognition. 2019.
>
> [5] Cheng, Weiyu, et al. "A neural attention model for urban air quality inference: Learning the weights of monitoring stations." Proceedings of the AAAI Conference on Artificial Intelligence. 2018.
>
> [6] Li, Yuanpeng, et al. "Weather forecasting using ensemble of spatial-temporal attention network and multi-layer perceptron." Asia-Pacific Journal of Atmospheric Sciences. 2021.

---

> ### Author Response · Authors · 2022-08-08
> **Response to Reviewer 3JHx**
>
> Dear reviewer 3JHx:
>
> We thank you for the precious review time and valuable comments. We have provided corresponding responses and results, which we believe have covered your concerns. We hope to further discuss with you whether or not your concerns have been addressed. Please let us know if you still have any unclear parts of our work.
>
> Best,
>
> NeurIPS 2022 Conference Paper312 Authors

---

### Official Review · Reviewer_7aqp · 2022-07-11

**Rating:** 6
**Confidence:** 3
**Soundness:** 3 good
**Presentation:** 3 good
**Contribution:** 3 good

**Summary:**

This paper presents a novel adversarial attack for spatiotemporal traffic forecasting models. Moreover, theoretically analysis are conducted to demonstrate the worst performance bound of the attack. Comprehensive experiments on real-world datasets show the effectiveness of the effectiveness of the attack, and how the robustness of the models enhances when combined with the corresponding adversarial training.

**Questions:**

See weakness.

**Limitations:**

Yes for limitations and no for potential negative societal impact.

**Strengths And Weaknesses:**

Strengths:
- The paper is well-written and easy to understand.
- Introducing adversarial robustness to spatiotemporal models is interesting and novel.
- The theoretical analysis backs up the proposed attack well.
- The experiments are comprehensive.

Weakness:
- I don't see clear weakness of the paper, although I wonder the generalizability of the method to other spatiotemporal-based applications beyond traffic forecasting models.

---

> ### Author Response · Authors · 2022-08-02
> **Response to Reviewer 7aqp**
>
> We are encouraged that the reviewer found our motivation and idea to be interesting and novel. Thanks for your positive opinions on the methodology and the insightful comments.
>
>
> ***
> > [W1] “I don't see clear weakness of the paper, although I wonder the generalizability of the method to other spatiotemporal-based applications beyond traffic forecasting models.”
>
>
> [Response] Thanks for the insightful comments. Indeed, our framework can be generalized to attack other spatiotemporal tasks based on geo-distributed data sources, such as air quality prediction [1] and weather prediction [2]. Similar to attacking traffic forecasting systems, we can fool the whole system by adding time-dependent adversarial perturbations to a few geospatially distributed monitoring stations. We will try to extend our framework to more spatiotemporal applications in the future.
>
> References:
>
> [1] Cheng, Weiyu, et al. "A neural attention model for urban air quality inference: Learning the weights of monitoring stations." Proceedings of the AAAI Conference on Artificial Intelligence. 2018.
> [2] Li, Yuanpeng, et al. "Weather forecasting using ensemble of spatial-temporal attention network and multi-layer perceptron." Asia-Pacific Journal of Atmospheric Sciences. 2021.
> ***

---

> ### Author Response · Authors · 2022-08-08
> **Response to Reviewer 7aqp**
>
> Dear reviewer 7aqp:
>
> We thank you for the precious review time and valuable comments. We have provided corresponding responses and results, which we believe have covered your concerns. We hope to further discuss with you whether or not your concerns have been addressed. Please let us know if you still have any unclear parts of our work.
>
> Best,
>
> NeurIPS 2022 Conference Paper312 Authors

---

### Meta-Review · Area_Chair_VGSA · 2022-08-27

**Recommendation:** Accept
**Confidence:** Less certain

**Metareview:**

The authors have made a significant effort to address reviewer concerns in their rebuttal. They are strongly encouraged to include these additional results and observations to either the main body of the paper or the supplement.

**Award:**

No

---

### Decision · Program_Chairs · 2022-09-14

Accept